# Silver Nanoparticles Loaded on Chitosan-g-PVA Hydrogel for the Wound-Healing Applications

**DOI:** 10.3390/molecules28073241

**Published:** 2023-04-05

**Authors:** Fahad M. Aldakheel, Dalia Mohsen, Marwa M. El Sayed, Khaled Ali Alawam, AbdulKarim S. Binshaya, Shatha A. Alduraywish

**Affiliations:** 1Department of Clinical Laboratory Sciences, College of Applied Medical Sciences, King Saud University, Riyadh 11433, Saudi Arabia; 2Clinical Laboratory Sciences Program, Inaya Medical College, Riyadh 12211, Saudi Arabia; 3National Research Centre, Giza 12622, Egypt; 4Chemical Engineering and Pilot Plant Department, National Research Centre, Giza 12622, Egypt; 5Respiratory Therapy Program, Inaya Medical Colleges, Riyadh 12211, Saudi Arabia; 6Department of Medical Laboratory Sciences, College of Applied Medical Sciences, Prince Sultan University, Al Kharj 16242, Saudi Arabia; 7Department of Family and Community Medicine, College of Medicine, King Saud University, Riyadh 4545, Saudi Arabia

**Keywords:** antimicrobial activity, chitosan, hydrogel, microwave irradiation, silver nanoparticles, polyvinyl alcohol

## Abstract

Silver nanoparticle composites have abundant biomedical applications due to their unique antibacterial properties. In the current work, green tea leaf extract was used as a natural reducing agent to synthesize AgNPs (AgNPs) using microwave irradiation technology. Furthermore, microwave irradiation has been used for the preparation of AgNPs/chitosan (Ch) grafted polyvinyl alcohol (PVA) hydrogel samples. To approve the accomplishment of AgNPs hydrogel polymer, UV-spectrum, TEM, and FT-IR spectrum analyses and the release of silver ions, actions were taken. The wound-healing ability of the prepared hydrogel samples was measured via both the in vitro (fibroblast cells) and the in vivo using rat models. It was found that chitosan-grafted polyvinyl alcohol, including AgNPs, exhibited excellent antibacterial activity against *E. coli* and *S. aureus* using the agar diffusion method. It can be said that microwave irradiation was successful in creating a hydrogel that contained silver nanoparticles. A wound that was still open was successfully treated with these composites.

## 1. Introduction

Antibacterial hydrogel is widely used in biomedicine, intelligent textiles, cosmetics, and many other fields. It is particularly used for medical applications because it has excellent antibacterial properties as well as good biocompatibility, water absorption and retention, and high oxygen permeability. The antibacterial hydrogel’s benefits as a wound dressing include its ability to absorb wound liquids, regulate medicine release, have no side effects, and may prevent secondary wound damage [1]. Nowadays, nanomaterials are acting as antibiotics, antifungals, and antivirals [2]. Nanoparticles provide several advantages over standard drug delivery strategies and medical applications as they enhance the solubility of hydrophobic medicines [2]. Antibacterial nanoparticles are employed in a variety of biological applications, including imaging, medication, gene delivery, and pathogen isolation [3].

AgNPs are used in a range of medical and biomedical fields as well as engineering and environmental applications [4]. They have been utilized in antibacterial cleaning, sterilization, and disinfection detergents for medical equipment, as well as a medication delivery carrier system [2,5]. These advantages can be correlated to their high stability, antibacterial capabilities, and ease of surface functionalization. AgNPs have been made using a variety of processes, including chemical reduction [6], lasers, and others [7]. Green tea extract is used around the world as a reducing agent and stabilizer for AgNPs and other elements [2,8]. The production of synthetic green AgNPs with good stability has progressed to the point that they are already being used in practical applications. As a result, the synthesis of green AgNPs became a very crucial issue in different applications. The phytochemicals in tea have a dual function, acting both as stabilizers to provide nanoparticle surfaces with a stable coating and as an active reducing agent to reduce gold, silver, and palladium in a one-pot procedure [9]. Several studies have demonstrated that the leaves and buds of the tea plant (Camellia sinensis), which is used to make tea, contain high concentrations of antioxidant polyphenols such as flavonoids and catechins [10,11].

Hydrogel is a 3D network with an excellent water absorption characteristic which includes a high-swelling ratio (500–900%) and quick equilibrium. In tests using the Gram-negative bacterium, e.g., *Escherichia coli*, and the Gram-positive bacteria, e.g., *Staphylococcus aureus*, the hydrogels also demonstrated strong antibacterial action. Therefore, a method for the environmentally friendly creation of antibacterial hydrogels based on AgNPs produced locally and conveniently from tea leaves was developed [2,8]. 

Chitosan is one of the most commonly used polysaccharides in wound-healing applications. When chitosan (Ch) is used without the addition of other compounds, it loses its strength quickly after absorbing wound exudate, shortening the time it takes to decompose. Hence, chitosan should be employed as a platform and cross-linked with synthetic polymers, such as polyvinyl alcohol (PVA) [12], to improve its mechanical strength and, hence, expand its application area. Furthermore, according to their antibacterial properties, AgNPs added to chitosan have an extremely efficient antimicrobial efficiency [13]. It extremely expands the surface area by allowing a further effective interaction with bacteria. Additionally, AgNPs-loaded hydrogel demonstrated remarkably effective antibacterial properties against *E. coli*, *P. aeruginosa*, *B. subtilis*, and *S. aureus*. This hydrogel released AgNPs gradually and continuously for at least seven days. Chitosan PEG hydrogel with AgNPs successfully supported the healing of diabetic lesions [14]. This composite has been applied in a variety of applications, including wound healing [15,16].

Numerous investigations have thus far supported AgNPs’ potent antibacterial action. *Staphylococcus aureus*, *Pseudomonas aeruginosa*, *Escherichia coli*, *Bacillus subtilis*, *Vibrio cholera*, *Salmonella typhus*, *Enterococcus faecalis*, *Klebsiella* sp., *Listeria* sp., and *Acinetobacter* sp. have all been found to be killed by AgNPs. AgNPs’ antibacterial mechanism is still up for debate. However, its antibacterial mechanism can be broken down into six categories: inhibition of the bacterial respiratory chain interference with protein synthesis and folding induction of bacterial genetic toxicity; induction of photocatalytic protein destruction and cell membrane rupture; and induction of the bacterial oxidative stress reaction to produce reactive oxygen species [17]; the mechanism of the silver nanoparticles’ antibacterial mechanism has been illustrated in Figure 1. Bacterial cell membranes can be destroyed by silver ions, which can also stop the respiration of bacteria. According to a study, silver has a stronger bactericidal effect in the nanoscale state because it may connect with the DNA in the cell membrane and stop the replication of the genetic material in bacterial cells [1]. Gram-negative bacteria were resistant to the AgNPs (15–25 nm; MIC: 16–128 g/mL), but not Gram-positive strains (15–25 nm; MIC: 256 g/mL) [18]. It is significant to note that, except for vancomycin against anti-microbial resistant bacterial strains, co-exposure to combinations of AgNPs and antimicrobial drugs, such as kanamycin, colistin, rifampicin, and vancomycin, demonstrated synergy against both wild-type and anti-microbial resistant *K. pneumoniae* isolates [19,20].

Consequently, hydrogels containing AgNPs and chitosan-grafted polyvinyl alcohol have been created employing various methods to improve their antibacterial properties and aid in wound healing [14]. These characteristics indicate that chitosan can kill bacteria, whereas AgNPs can disseminate and destroy germs in the environment [18,19].

Many studies have recommended employing hydrogels loaded with nanoparticles made from different biocompatible polymers, such as chitosan and alginate, to treat skin-wound healing [10]. AgNPs have been prepared with chitosan and a polyvinyl alcohol coating that showed good antibacterial and biocompatibility qualities [20]. A gelatin/carboxymethyl chitosan matrix has been created and cured with silver nanoparticles. This matrix possesses exceptional physical properties as well as significant antibacterial properties [21]. Oxygen plasma has been used to load chitosan/polyenyl alcohol gels after modifying poly L-lactic acid with silver nanoparticles. There was biocompatibility, as well as antibacterial activity [22,23,24]. The current study will use green tea leaves as a green-reducing agent to biosynthesize AgNPs after loading them into chitosan hydrogels using a microwave irradiation technique [25,26]. The antibacterial action against *Gram-positive* and *Gram-negative* bacteria of the generated hydrogels has been studied. Transmission, electron microscopy (TEM), and Fourier transform infrared (FTIR) spectroscopy were used to characterize the synthesized hydrogels. The antimicrobial activity of hydrogels has been investigated as a function of silver nanoparticle concentration. Their antibacterial properties can be used topically to treat infections and promote wound healing.

## 2. Results and Discussion

The AgNPs used in this work were produced with green tea (Xinyang Maojian Tea) and were water stable.

### 2.1. Characterization of the Prepared AgNPs

In all of the investigated samples, AgNPs had a spherical shape, as seen in Figure 2. TEM was used to determine the size of AgNPs. Figure 2A,B revealed that the sizes of S1Ag0.3 ranged from 3 to 16 nm, whereas Figure 2C,D showed that the sizes of S1Ag0.8 ranged from 4 to 19 nm. The amount of AgNPs in S1Ag0.8 hydrogel was higher than the amount of S1Ag0.8 hydrogel, according to the results. The hydrodynamic diameter of the AgNPs is determined by the DLS technique (Figure 2B). According to DLS analysis, this size distribution of AgNPs is centered at 22.31 nm.

In addition, as shown in Figure 3, according to FTIR data, the spectrum of AgNPs loaded in S1Ag0.3 and S1Ag0.8 showed an absorption peak at 428 nm, which was due to the AgNP-localized surface plasmon resonance [20]. In S1Ag0.3 hydrogel, a broad absorption peak of AgNPs was formed at low AgNP concentration, whereas S1Ag0.8 revealed that increasing AgNP concentration resulted in more acute absorption. The distribution of AgNPs in S1Ag0.8 hydrogel was higher than in S1Ag0.3 hydrogel [22].

### 2.2. Hydrogel Antimicrobial Activity

The agar diffusion method was used to assess the hydrogel polymer swelling behavior as well as the antibacterial characteristics of the hydrogel polymer with various concentrations of silver ions.

Pathogens distributed in water should interact with the hydrogel antibacterial nanocomposite. A hydrogel’s ability to swell is, therefore, a crucial component of its use. The hydrogel nanocomposite absorbed water quickly in the first 20 min before gradually reaching equilibrium, demonstrating the high hydrophilicity of the porous structure (Figure 4). In terms of swelling behavior, samples S1Ag0.3 and S1Ag0.8 hydrogels performed well. Because of the decreased water solubility, AgNPs may lower the swelling ratio, and the presence of more AgNPs also decreased the crosslink density. The silver-based nanocomposite gel’s swelling capacity and antibacterial properties were both impacted by the variable surface charge caused by the varying AgNP content. These biomaterials’ antibacterial properties essentially rely on the silver ions produced from the hydrogels. As shown in Figure 4, the quantity of silver ions released from S1Ag0.3 and S1Ag0.8 hydrogels by utilizing the F12 medium with regard to time was measured by AAS. The amount of silver ions released from S1Ag0.3 hydrogels was 5.9 g/mL within the first 6 h and quickly climbed to 13 μg/mL over the next 7 h. The release rate increased from 12 to 24 h, reaching 18 μg/mL. After the first 24 h, the release rate increased somewhat, reaching a value of 19 g/mL at 48 h. In comparison to the findings from the S1Ag0.3 gel, the amount of silver released on the S1Ag0.8 gel was twice as large. The inhibitory zone grew as the number of AgNPs increased, according to the provided data.

The inhibition zones diameter proves that both Gram-negative strains and Gram-positive were significantly inhibited for the four samples S1, S1Ag, S1Ag0.3, and S1Ag0.8. Moreover, Gram-negative *E. coli* was more susceptible to being inhibited by using the topical hydrogel with AgNPs than *S. aureus*. Nearly similar results have been confirmed with reference [27].

The results of the MTT assay, as shown in Figure 5, were used to detect the toxicity of S1Ag0.3 and S1Ag0.8. The results of the tests on the viability of the L-929 fibroblasts and the extracted solutions of S1, S1Ag0.3 hydrogel, and S1Ag0.8 hydrogel demonstrated that the increase in AgNPs, and AgNP concentrations in hydrogels led to a decrease in the hydrogels’ and cells’ viability. Definitely, at the same concentration of 100% extracted solution, polyvinyl only had about 90% cell viability, while S1Ag0.3 and S1Ag0.8 had approximately 70% and 80% cell viability, respectively. Nevertheless, all samples’ cell viability was higher than 70%, which was the threshold between cytotoxicity and non-cytotoxicity, so they can be considered non-cytotoxic. Furthermore, images of fibroblast cells proliferating on a 100% extracted solution of hydrogels revealed that the number of fibroblast cells proliferating on the extracted solution was greater than that of S1Ag0.3 and S1Ag0.8. An excessive concentration of AgNPs may have a negative impact on skin and wound healing. Our results agreed with those mentioned by [25,26,27]. The tested cell viability of L-929 cells with the extracted solutions of S1, S1Ag0.3, and S1Ag0.8 showed that the increase in AgNP amount in gels resulted in a decrease in cell viability and had a bad effect on skin wound healing. In addition, in this study, the antimicrobial properties and biocompatibility of hydrogels were assessed.

### 2.3. In Vivo Skin-Wound Healing

In vivo wound-healing capacity was investigated through the detection of morphological changes in wound reduction size, as appeared in Figure 6. The results illustrated in Figure 7 demonstrated the differences in skin wound size during the wound-healing phase of each sample, which was used to calculate the wound size decrease. There was a non-treated control group related to S1 group wounds that healed more than those in the other groups. However, because the measurement was relative, the difference between the S1, S1Ag0.3, S1Ag0.8, and the S1 group was not significant. On day 7, all of the wounds, which ranged in size from 23% to 30%, had a scab formation over the defective area, which prevented accurate measurement of the wound size. Day 8 reveals partial wound closure in all groups, ranging from 60 (3%) to 75 (11%). The mice were put to death on day 12, and the scabs from their wounds were removed once more. It revealed that up to 98 to 99 % of the wounds had healed. Animals treated with S1Ag hydrogel experienced greater wound size reduction than those treated with S1; however, among the hydrogels, S1Ag0.3 and S1Ag0.8 were deemed more effective. S1Ag0.8 wounds were highly specific. Finally, when the wound-healing process had progressed to remodeling and no more scabs remained on the wounds on day 15, the post-implantations were removed and their histological structures examined, as shown in Figure 7. S1Ag0.8 showed better wound healing than the control and S1Ag0.3, according to the histological investigation. Six days after fixing, inflammatory cells and newly formed fibrous tissue were discovered in both the control and S1 samples. Even though the wounds had been treated, they were totally covered in the de-epithelialized epidermis and newly produced fibrous tissues. In addition, as offered in Figure 8, the spectrum of AgNPs loaded in S1Ag0.3 and S1Ag0.8 showed an absorption peak at 428 nm due to the silver nanoparticles’ localized surface plasmon resonance [25]. In S1Ag0.3 hydrogel, a broad absorption peak of AgNPs was formed at low AgNPs concentration, whereas S1Ag0.8 revealed that increasing AgNPs concentration resulted in more acute absorption. The distribution of AgNPs in S1Ag0.8 hydrogel was higher than that in S1Ag0.3 hydrogel.

The bar graph’s asterisk denotes (*p* ≤ 0.05) Significant difference.

Dressings are necessary for wound healing and infection prevention. An antimicrobial mediator is required in a good wound dressing. An antibacterial comprising AgNPs could be produced in a hydrogel polymer. This polymer is used to control the release of silver (Ag) ions as well as to extend AgNPs’ antibacterial activity. Biodegradation, biocompatibility, and absorption of extra liquid material are all requirements for the hydrogel polymer. As a result, combinations of AgNPs and hydrogel synthetic polymers have been investigated in order to achieve both benefits. The combination of polyvinyl alcohol (PVA) and chitosan (S1) has been created as a reducer and template for controlling AgNP release. Different methods can be used to mix the produced AgNPs into hydrogel polymers, such as the multicomponent AgNPs–chitosan polyvinyl alcohol electro-spun for wound dressing applications described [17]. An electro-spun membrane has been established that contains AgNPs-PVA-COS and displayed exceptional antibacterial properties and biocompatibility [16]. In our study, cross-linkage chitosan–polyvinyl alcohol to reduce Ag^+^ to Ag^0^ using microwave irradiation. The advantage of this method is that the hydrogel polymer is made up without using any reducer. Hence, there is no need to further remove residual crosslink and reducer.

The results showed that TEM is a tool generally used to characterize the surface and size of the synthesized silver nanoparticles. The TEM results displayed that AgNPs of different sizes were synthesized by the reduction of AgNPs with the green tea extract. At low magnifications, highly polydisperse, large-size AgNPs were observed. It was apparent from the TEM image that the AgNPs had a distinct uniform interparticle separation from each other. The shape of the nanoparticles prepared by chemical reduction can be affected by the molecular structure of the reductive agent. The presence of reductive agents in the extract of green tea led to the formation of flower-shaped nanoparticles. Such an observed shape of the prepared nanoparticles was supportive proof that could help explain their antimicrobial activity. Furthermore, as the concentration of AgNPs is increased, the rate of AgNPs larger than 10 nm increases, although aggregation appears to be slightly reduced and the average diameter increases. The creation of silver nanoparticles, AgNPs, was visually confirmed by the change in color of the hydrogels and the UV spectrum. The color of white gels changed to deep brown after being microwaved, indicating the presence of silver in nanoparticle production [15,28,29,30]. In the UV-vis spectrum of the silver nanocomposite hydrogel shown in Figure 2, the S1Ag0.3 hydrogel showed a broad absorption peak of AgNPs at a low concentration of AgNPs, whereas S1Ag0.8 demonstrated that a higher concentration of AgNPs resulted in stronger sharp absorption. Therefore, results showed that the prepared hydrogels were successfully fabricated by using microwave assistance, as indicated by TEM observation. For medical applications, the biocompatibility of S1 gels is another important factor that needs to be confirmed.

Silver ion release rates were higher in S1Ag0.3 than in S1Ag0.8 for nearly 24 h. After 24 h, no significant change was observed. This implies that the concentration of AgNPs should be higher in order to claim active antibacterial activity. The hydrogel polymer nanocomposite antimicrobial activity of each sample for Gram-negative strains and Gram-positive strains (*E. coli* and *S. aureus*) is shown in Figure 5. The inhibition zone diameter for all samples demonstrates that both Gram-negative and Gram-positive strains were significantly inhibited. The silver release rate from S1Ag0.3 and S1Ag0.8 hydrogels was determined using the F12 medium and AAS. For S1Ag0.3 hydrogels, the amount of silver ions released was 5.9 μg/mL within the first 6 h and quickly increased to 13 μg/mL for the next 7 h. The release rate increased from 12 to 24 h, reaching 18 μg/mL. The release rate then increased slightly after the first 24 h, reaching 19 μg/mL after 48 h. For S1Ag0.8 gel, the silver amount release was twofold in comparison with the data for S1Ag0.3 gel. Laterally, the results revealed that, as the amount of AgNPs increased, the inhibition zone increased. The current result was agreed upon by some previous research [26]. Briefly, the release rates of silver ions from S1Ag0.8 and S1Ag0.3 were 13.5, 25.4, 32.3, 33.2, 34.2, 35.5, and 38.4 for 1, 6, 12, 24, 48, 168, and 338 h, respectively. The same results were illustrated by Andrews’ determination that the antibacterial activity against *E. coli* and *S. aureus* was affected by the concentration of the released silver ions. On the contrary, the inhibition zone was not detected in S1 and S1Ag samples for both Gram-negative bacteria and Gram-positive bacteria [24]. The inhibition zone diameter of each sample produces significance with *p* ≤ 0.05 statistically with the control group S1. The diameter of the inhibition zone increased as the concentration of silver ions increased from 0.5 to 1 ppm. The diameter of the inhibition zone for *E. coli*, when treated with S1Ag0.8, is clearly 184 mm, which is 0.4 mm larger in diameter than S1Ag0.3. In the current study [31], the authors discovered that the inhibition diameter of each sample produces statistically significant results, i.e., *p* ≤ 0.05 when compared to the control group. When we doubled the concentration of silver ions from 15 ppm to 30 ppm, we observed an increase in inhibition diameter. Clearly, *S. aureus* (a Gram-positive organism) is inhibited with a 60 (3 mm) diameter when treated with the gel PCA30, 0.2 mm diameter more than the gel PCA15, and the gel PCA60 retains that inhibition zone against *S. aureus*. Meanwhile, the diameter of the inhibition zone for *S. aureus* is 163 mm, as measured with the S1Ag0.8 hydrogel, which is larger than 0.3 mm when compared to the S1Ag0.3 hydrogel. The results in Figure 4 declared that the S1 did not generate an inhibition zone in both Gram-positive and negative bacteria. A similar antimicrobial property for S1Ag0.3 and S1Ag0.8 gels had been shown against *E. coli*, while an increase in antimicrobial activity against *S. aureus* was observed as the concentration of AgNPs increased S1Ag0.8 had better antimicrobial activity against *S. aureus* than S1Ag0.3. Figure 4 illustrates that both S1Ag0.3 and S1Ag0.8 did not generate toxicity in cell culture tests. The results of the current study were agreed upon [3]. They mentioned that a higher concentration of AgNPs produces an excellent antimicrobial effect. As shown in Figure 5, the creation of a thick scab on the wound suggested healing in the control (S1), S1Ag0.3, and S1Ag0.8, but it was thinner in the case of chitosan and polyvinyl hydrogel (S1). The thick skin scabs were separated, and the wound size decreased adequately in 85 percent of all samples when compared to the initial wound size on day 12. In addition, wounds treated with S1Ag0.8 did not develop a skin scab. The ability of AgNPs chitosan polyvinyl alcohol has been demonstrated to promote wound healing in the PCA30 sample in vivo. Therefore, the hydrogel polyvinyl alcohol/chitosan loaded with AgNPs has a potential application as an antibacterial topical gel [20,30]. Lastly, according to the histological investigation, the image of S1Ag0.8 displayed that the gel could decrease the inflammation of the skin and induce a skin-healing process better than S1Ag0.3 does. The present work agreed with that described by [3] as the H&E staining of sections of wounds treated by high concentrations of a hydrogel loaded with high concentrations of AgNPs that decrease the inflammation of the skin and produce wound healing better than hydrogel loaded with low concentrations of AgNPs [20]. The results of the existing study could produce better antibacterial effects by using different concentrations of AgNPs loaded into hydrogel polymers. Overall, the in vivo tests shown in Figure 6, Figure 7, and Figure 8 revealed that both S1Ag0.3 and S1Ag0.8 could reduce wound skin inflammation, support the formation of novel fibrous tissue, and completely epithelialize the epidermis. 

## 3. Methods

### 3.1. Ethical Considerations

Research protocols for animal injection were approved by the National Research Centre (Animal Facility Unit). The date of the experiment of this study was 17 March 2022, under number (RSP 2023R506) from KSU.

### 3.2. Experimental Animals and Housing Conditions

All experimental and animal care procedures were carried out in accordance with the international guidelines governing the animal care and use committee’s recommendations, as well as international rules governing the care and use of laboratory animals. As a result, 80 Sprague-Dawley rats (weighing 200–250 g) were obtained from a nearby provider, and the animals were gathered in groups. The rats were housed in group cages with temperature control (23–25 °C), humidity control (60%), and 12-h light/dark cycles. To urge the rats to eat less, they were given traditional, pelleted food and water. The rats were free of infections when they arrived at the workplace. *E. coli* and *S. aureus* were used in the antibacterial tests. Rats were sacrificed via an anesthetic overdose of Dimethyl ether.

### 3.3. Test Strains/Tissue Cell Line

Test strains were obtained from Microbiology and Biotechnology Department at National Research Center (NRC). The VITEK 2^®^ version 9.02 system was used to identify *E. coli* and *Staphylococcus aureus* (*S. aureus*), which were then kept at 80 °C in tryptic soy broth (Oxoid, England) with 20% glycerol. Bacteria were distributed over sheep blood agar and incubated aerobically at 37 °C for 24 h for propagation. Colonies were suspended in tryptic soy broth (TSB); strains were cultured till they reached the logarithmic growth phase and then standardized using the EUCAST method. Chitosan with a medium molecular weight (400,000 g/mol, 75–85 percent units deacetylate; shrimp origin) and 99% hydrolyzed PVA (MW 124–146 kg/mol) was purchased from Sigma–Aldrich. ATCC provided the fibroblast cell line (L-929) for the in vitro study. As initiators and cross-linkers, the researchers used potassium persulfate (KPS) (Merck, Munich, Germany), methylene, and bis-acrylamide (MBA) (Fluka, Munich, Germany). 

Human dermal fibroblasts were obtained from Zen-BIO and shipped in dry ice, and the catalog number was (1e6cells).

### 3.4. Green’s Synthesis Method for AgNPs 

A method of making AgNPs is sustainable because green tea contains flavonoids and catechins. By boiling 100 g green tea leaves (Herbarium No. 304) in 50 mL of distilled water for 30 min, the green tea leaf extract was properly prepared. To obtain a clear extract, the tea filtrate was centrifuged at 5000 rpm for 10 min. A total of 10 mL of tea extract was thoroughly mixed with 1 mL of 0.1 M AgNPs and 10 mL of water. At 25 °C, the mixture was stirred at 700 rpm. The mixture’s color changed from light brown to green, indicating the synthesis of AgNPs [1]. To repeat the operation, several volumes (15 mL, 10 mL, 5 mL, and 3 mL) of tea extract were employed, as shown in Table 1. From 1 to 4, the biosynthesized AgNPs were all referred to as AgNPs.

### 3.5. Synthesis of Chitosan Grafted PVA/AgNPs Hydrogels

Hydrogel samples were synthesized as described [24] with some changes, including using the microwave technique in the grafting process. Briefly, we mixed 1 g of KPS and 0.2 g of MBA as initiator and cross-linker, respectively, and 2% *w*/*v* chitosan (in acetic acid) with 10% *w*/*v* solutions of polyvinyl alcohol (PVA) in an equal volume. This step was followed by dividing the resulting sample into four parts: the first portion is the control sample (S1). The second part (S1Ag) has been developed by dissolving silver nitrate nanoparticles and powder in water to get 10% *w*/*v* AgNPs, which have been mixed well. The amounts of 0.3% *w*/*v* and 0.8% *w*/*v* of the prepared AgNPs were added to the third and fourth samples (S1Ag0.3 and S1Ag0.8), respectively. Then, the four samples were irradiated in a Hitachi, Tokyo, Japan, microwave oven at 800 W for 10 min. A simplified demonstration of the hydrogel synthesis process is clarified in Figure 9. Finally, for both in vitro and in vivo tests, the samples produced were sterilized using UV irradiation.

### 3.6. Release of Silver Ions

Atomic absorption spectrometry (AAS) was used to investigate the rate of release of the created silver ions by measuring numerous silver solutions with different silver ion concentrations [29].

### 3.7. Characterization Using TEM and FTIR

The transmission electron microscopy (TEM) technique was utilized to characterize AgNPs and determine the average size of the AgNPs. A Tecnai G2 F20 microscope was adjusted at 200 kV to examine the form and size of AgNPs embedded in chitosan/polyvinyl alcohol hydrogel. AgNPs (0.1 gm) were dispersed in 5 mL of distilled water. A drop of the particle suspension was placed on a carbon-coated copper mesh and allowed to dry at room temperature. The resulting images were used to determine the average size of the AgNPs. PVA and chitosan interactions were revealed by FTIR spectra obtained from an FTIR analyzer device (PerkinElmer Spectrum GX, Los Angeles, CA, USA).

### 3.8. Statistical Analysis

SPSS (version 19) was used to conduct the statistical analysis. The mean and standard deviation of all data is available (S.D.). Statistical differences with a *p* ≤ 0.05 significance level were judged significant.

### 3.9. In Vitro Antimicrobial Assessment Using Agar Diffusion Test

Gram-negative *E. coli* (ATCC 25922) and Gram-positive *S. aureus* (ATCC 6538) were employed as model microorganisms. Overnight single *E. coli* and *S. aureus* colonies were plated and transferred to tubes with sterilized liquid LB (Luria Bertani) to create a seed culture that was grown at 37 °C, diluting the developing culture into flasks containing fresh LB medium, and the flasks were maintained in the same culture conditions. The optical density at 600 nm has been used to track the growth (OD600). With sterile glass beads, the cell suspension (50 L) was equally dispersed onto LB agar (90 mm). To ensure that the hydrogels employed in the antimicrobial activity assay were equivalent in weight and morphology, the four samples produced were used to coat the plate’s surface in a mold with the same diameter (1.5 cm). After 24 h of incubation at 37 °C, the zone of inhibition was assessed.

An atomic absorption spectrometer (AAS) was used to quantify the concentration of silver ions released at a certain rate (Z5000, Hitachi, Tokyo, Japan) via atomic absorption spectrometry measurement of several silver solutions with a given silver ion concentration (AAS, Shanghai, China). After that, the hydrogel was immersed in a 50-mL acetate buffer (pH 5.5). The samples were incubated at 37 °C at 100 rpm stirring. At defined intervals, aliquots of each sample (1 mL) were obtained from the release medium and diluted to 5 mL with a new buffer solution to determine the amount of medication released at different times up to five days. After overnight incubation of the inoculated plates upside down at 37 °C, the zones with the lowest inhibitory concentrations were studied. The test was repeated three times with the same bacterial strain. Approximately 15 mg of S1, S1Ag, S1Ag0.3, and S1Ag0.8 were added into 50 mL of bacteria nutrient solution with a bacterial concentration of 10^4^ colony-forming units per ml. A bacteria-nutrient solution with the same concentration was used as the control. Figure 10 shows the disc sensitivity test for *E. coli* (A, C, E, and D) and *S. aureus* (E, F, G, and H).

### 3.10. Human Skin Fibroblast Cells Cytotoxicity Evaluation

The cytotoxicity of human skin fibroblasts was investigated. UV light was used to sterilize the four prepared samples for one hour on both sides. The cells were then cultured for 24 h in a serum-free medium containing only Dulbecco’s Modified Eagle’s Medium, giving extraction media with concentrations of 0.2, 0.4, and 0.8 mg/mL. Filtration of the supernatant using a Millipore membrane with a pore size of 0.22 mm (Millipore Billerica, Burlington, MA, USA). As a negative control, 10% fetal bovine serum in Dulbecco’s medium was employed. Human skin fibroblasts were seeded at a density of 10^5^ cells per well in 96-well plates in 90 L of Dulbecco’s Modified Eagle’s Medium supplemented with 10% fetal bovine serum. The cells were cultured at 37 °C in a wet atmosphere containing 5% CO_2_. After 24 h, the extraction medium containing 10% fetal bovine serum was replaced, and the cells were incubated for an additional 24 h. The extraction solutions that had been tested were then removed. Finally, the cells were cultured for 4 h in 100 mL of MTT-containing media (1 mg/mL). To dissolve the formazan crystals, the media was discarded from the wells and replaced with 200 L of dimethyl sulfoxide (DMSO). Cell viability (%) was calculated based on the absorbance at 590 nm using an ELISA reader. The test was repeated three times, and cell viability was calculated as a percentage relative to the control.

### 3.11. In Vivo Study and Histological Examination

In this work, wound healing was determined using dispersion hydrogel on Dawley rate dorsal opening wounds, which was approved by the ethical committee. Dimethyl ether was used to anesthetize Dawley rats, who were then placed on a table with their backs shaved and then cleaned up with a povidone solution. Two full-thickness lesions with a diameter of 1 cm were produced on each dorsal of the rats, and 0.1 mL of gel was administered. The control group consisted of non-treated Dawley rats. After obtaining wound images for a 15-day period to assess wound healing [27], rats were murdered on day 15. Extracted samples were stored in a formaldehyde solution with a formaldehyde content of 10% by volume. The extracted samples were dried, then embedded in paraffin, and finally sectioned with a microtome to get 5 mm-thick slices. For histological analysis, the previously prepared sectioned slices were further stained with hematoxylin and eosin (H&E). A light microscope was used to view the stained slice samples (Nikon Eclipse, Ti-U series, Tokyo, Japan).

## 4. Conclusions

According to the current study, AgNPs with average diameters of 22.31 nm were successfully prepared using a green tea leaf. The prepared nanoparticles have been loaded on chitosan-g-polyvinyl alcohol antimicrobial hydrogels using microwave irradiation techniques. Both cross-linking and reduction actions for silver from Ag+ to Ag0 were produced through microwave irradiation. The hydrogel antimicrobial activity showed that the quantity of silver ions released from S1Ag0.3 and S1Ag0.8 hydrogels by utilizing the F12 medium with regard to time was measured by AAS. Furthermore, in vivo skin-wound healing on day 7 of all of the wounds ranged in size from 23 % to 30%. Therefore, results showed that the prepared hydrogels were successfully fabricated by using microwave assistance, as indicated by TEM observation. Silver ions’ release rate increased if the concentration of AgNPs increased. S1Ag0.3 and S1Ag0.8 were nontoxic and biocompatible and exhibited sustained-release characteristics, demonstrating excellent antibacterial activity against Gram-positive *S. aureus* and Gram-negative *E. coli.* Nevertheless, the antimicrobial action against *E. coli* and *S. aureus* was influenced by silver ion concentration. For medical applications, the biocompatibility of S1 gels is another important factor that needs to be confirmed. Lastly, according to the histological investigation, the image of S1Ag0.8 displayed that the gel could decrease the inflammation of the skin and induce a skin-healing process better than S1Ag0.3 does.

## Figures and Tables

**Figure 1 molecules-28-03241-f001:**
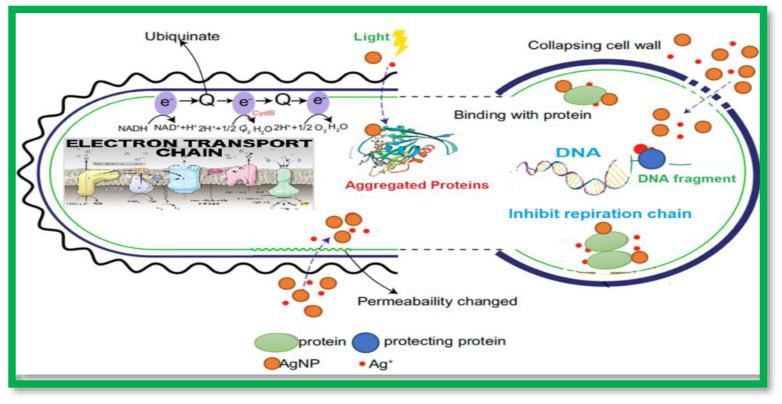
Silver nanoparticles’ (AgNPs) antibacterial mechanism.

**Figure 2 molecules-28-03241-f002:**
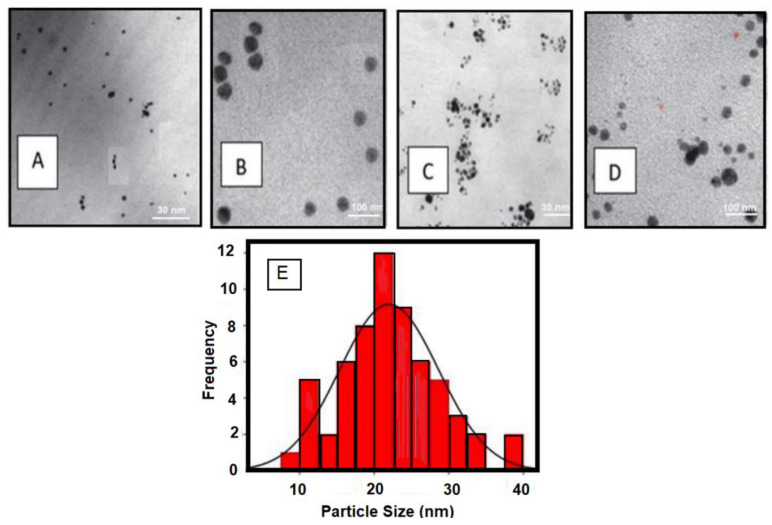
TEM for S1Ag0.3 (**A**,**B**) and S1Ag0.8 (**C**,**D**) at Magnification (30 k and 100 k) and (**E**) DLS of the Synthesized Silver Nanoparticles.

**Figure 3 molecules-28-03241-f003:**
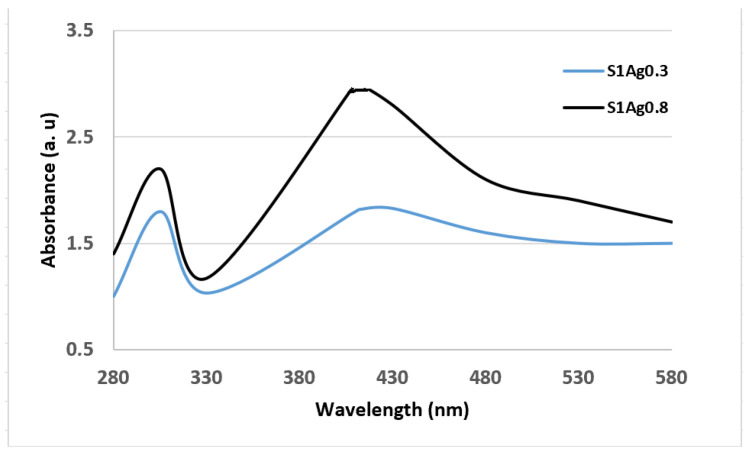
FTIR absorption spectra of S1Ag0.3 and of S1Ag0.8.

**Figure 4 molecules-28-03241-f004:**
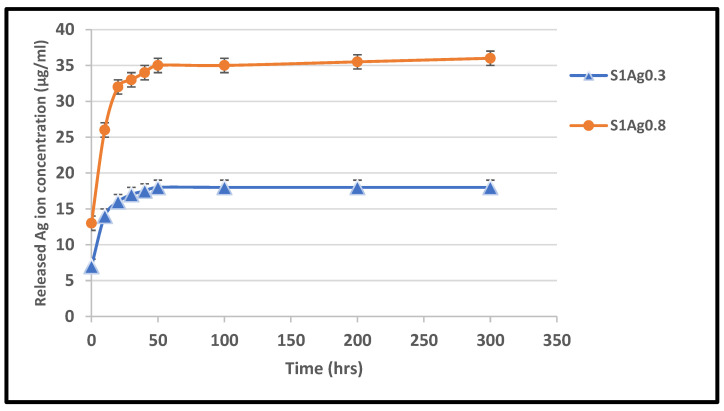
Silver release rates of S1Ag0.3 and S1Ag0.8 hydrogels.

**Figure 5 molecules-28-03241-f005:**
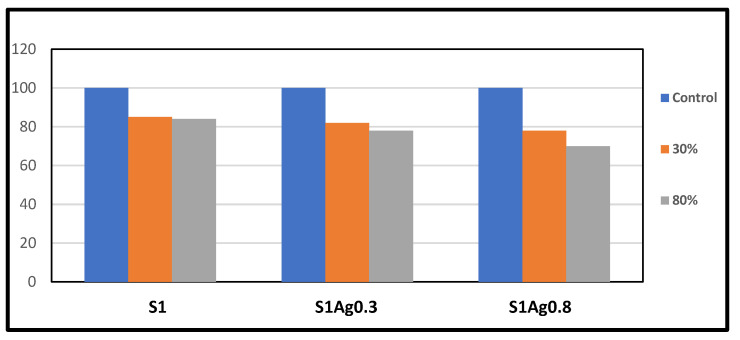
Cytotoxicity tests from MTT assay of the cell viability, detection of the toxicity of S1, S1Ag0.3, and S1Ag0.8. Absorbance was normalized to that of the negative control at each time interval and was considered 100%. The data are presented as the mean ± standard deviation (*n* = 5).

**Figure 6 molecules-28-03241-f006:**
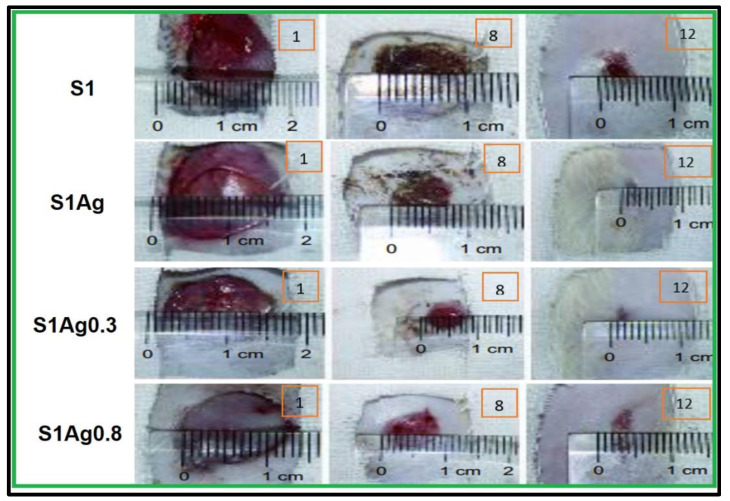
Wound size reduction rate at 1, 8, and 12 days after grafting with the four samples.

**Figure 7 molecules-28-03241-f007:**
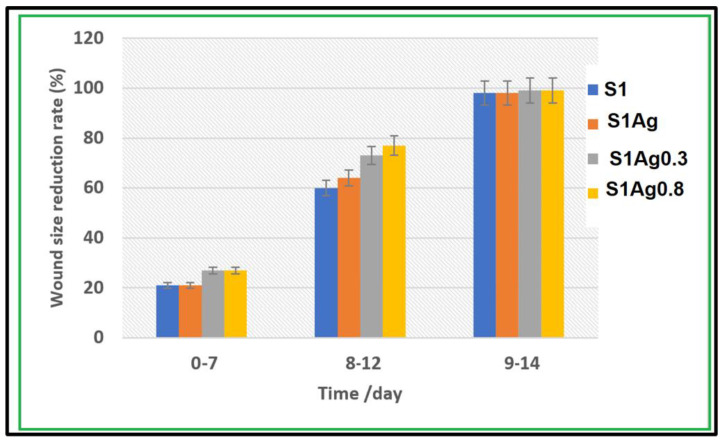
Skin wound size reduction after 7, 12, and 14 days.

**Figure 8 molecules-28-03241-f008:**
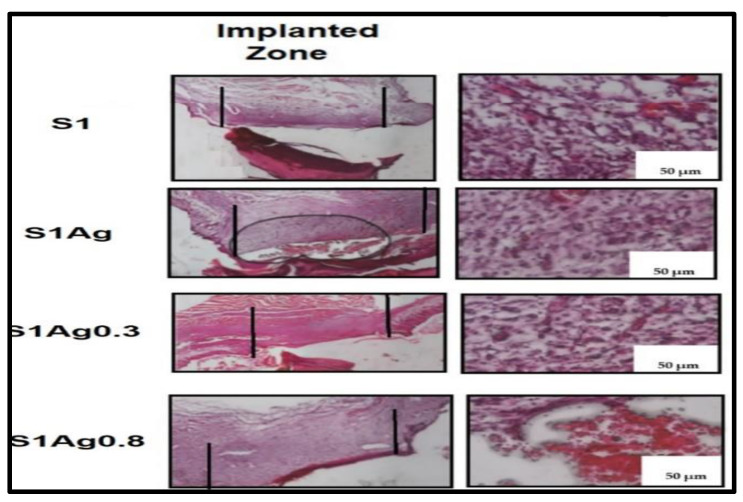
H&E staining images of full defected area of the skin wound treated (implanted zone) in the four samples after 15 days.

**Figure 9 molecules-28-03241-f009:**
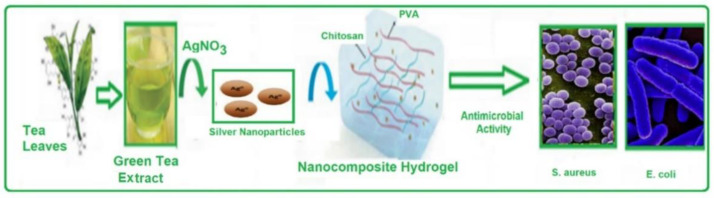
A schematic diagram showing the preparation of Ch-g-PVA-NPs hydrogels.

**Figure 10 molecules-28-03241-f010:**
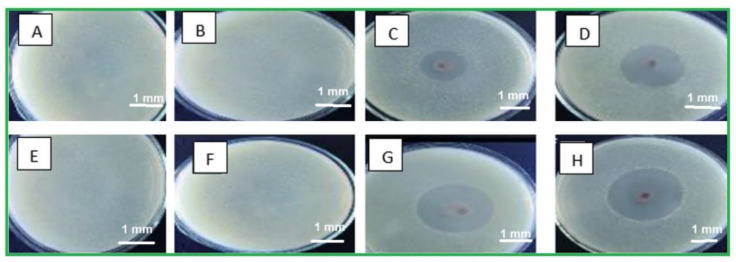
The inhibition zone for S1 (**A**,**E**), S1Ag (**B**,**F**), S1Ag0.3 (**C**,**G**), and S1Ag0.8 (**D**,**H**).

**Table 1 molecules-28-03241-t001:** Designated Tea Silver Nanoparticles (TAgNPs) (20 g/L).

No.	Green Tea Extract Volume “mL”	AgNPs (0.1 M) “mL”	Nanoparticle Diameter “nm”
1	15	1	Mean = 22.31 nm SD = 8.36 Counted = 32 particles
2	10	1	
3	5	1	
4	3	1	

## Data Availability

The data presented in this study are available on request from the corresponding author.

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
