# Peer review of "Silver Nanoparticles Loaded on Chitosan-g-PVA Hydrogel for the Wound-Healing Applications"

_molecules, 2023, doi:10.3390/molecules28073241_

Round 1

Reviewer 1 Report (Previous Reviewer 2)

The manuscript has been significantly improved and now warrants publication in Molecules. 

Author Response

Dear Reviewer

I did the required modifications according to your advice 

Thank you 

Reviewer 2 Report (Previous Reviewer 4)

It can be published in the present form. 

Author Response

Dear Reviewer

I did the required modifications according to your advice 

Thank you 

Reviewer 3 Report (Previous Reviewer 3)

The manuscript can be accepted for publication in its present form.

Author Response

Dear Reviewer

I did the required modifications according to your advice 

Thank you 

Reviewer 4 Report (New Reviewer)

This manuscript has lack of novelty. Therefore, I am rejecting this manuscript. 

Comments

Summary

This work reports the microwave supported synthesis of Silver nanoparticles/Chitosan/PVA hydrogel for in vivo wound healing application. I found a very similar article as mentioned below and surprisingly images of the reported antimicrobial activity of the submitted manuscript (Fig. 7-1,2,3,4,5,7) are probably same of the below mentioned reference article (Fig. 3- a,b,c,f,g,h). This might subject to further evaluation. 

Nguyen, T. D., Nguyen, T. T., Ly, K. L., Tran, A. H., Nguyen, T. T. N., Vo, M. T., ... & Nguyen, T. H. (2019). In vivo study of the antibacterial chitosan/polyvinyl alcohol loaded with silver nanoparticle hydrogel for wound healing applications. International Journal of Polymer Science2019.

I am not recommending this manuscript to be published in "Molecules".

Thanks

Author Response

Dear Reviewer

Kindly find the attached file.

Thank you 

Reviewer 5 Report (New Reviewer)

Comments to the Author
The authors have submitted the manuscript entitled “Microwave-Supported Synthesis of Silver Nanoparticles/Chitosan-Grafted-Polyvinyl Alcohol Hydrogel for Wound Healing Applications
” Some questions need to be answered.

1. First, in the introduction section, it's better to relate the types of methods with green tea extract.

2. In line 55, it was mentioned that "In tests using the gram-negative bacterium e.g. 54 Escherichia coli and the gram-positive bacteria e.g. Staphylococcus aureus the hydrogels also demonstrated strong antibacterial action”. But it was not determined what the composition of the hydrogen was. Antibacterial activity is not true for all types of the hydrogel.

3. In table 1, the last column, nanoparticle diameters were not calculated for all samples.

4. It is not clear whether the nanoparticles made from green tea were used in the hydrogel or not

5. Also, for example: in line 188, silver nitrate nanoparticles are mentioned, if this sentence is true, the synthesis of silver nanoparticles has not been done.

6. AgNO3 can dissolve, so the release of silver ions in the hydrogel can be natural. Unless silver nanoparticles are used.

7. In the results section, only TEM images are related to two samples while 4 samples have been synthesized. Also, DLS has only been reported for an unknown sample.

8. Also, samples S1Ag0.3 and S1Ag0.8 are related to hydrogel, but the discussion is about nanoparticles with different codes.

9. In lines 301 -308, it was discussed silver ion release, but it is very unclear. Also, only one sample has been discussed.

10. It recommended phase analysis of silver nanoparticles by XRD, and adding EDX for samples

Author Response

Dear Reviewer

Kindly find the attached file. I did the required modifications according to your advice 

Thank you 

Round 2

Reviewer 5 Report (New Reviewer)

The authors have submitted the revised manuscript entitled “Microwave-Supported Synthesis of Silver Nanoparticles/Chitosan-Grafted-Polyvinyl Alcohol Hydrogel for Wound Healing Applications” Some corrections need to be applied to the manuscript.

1-      Please use only notation and remove silver nanoparticles from sentences. (Such as Page 9 lines 296-297, Page 13 lines 385-386). Also, silver nitrate nanoparticles should be removed (Page 5 line 174).

2-      Please reconsider the conclusion based on the title and results. 

Author Response

The authors have submitted the revised manuscript entitled “Microwave-Supported Synthesis of Silver Nanoparticles/Chitosan-Grafted-Polyvinyl Alcohol Hydrogel for Wound Healing Applications” Some corrections need to be applied to the manuscript.

  • Please use only notation and remove silver nanoparticles from sentences. (Such as Page 9 lines 296-297, Page 13 lines 385-386). Also, silver nitrate nanoparticles should be removed (Page 5 line 174).

All “ silver nanoparticles” has been replaced by “AgNPs”

  • Please reconsider the conclusion based on the title and results. 

The conclusion has been modified completely.

This manuscript is a resubmission of an earlier submission. The following is a list of the peer review reports and author responses from that submission.

Round 1

Reviewer 1 Report

Manuscript: Microwave-Supported Synthesis of Silver Nanoparticles/Chitosan-Grafted-Polyvinyl Alcohol Hydrogel for Wound Healing Applications

In the Manuscript authors deals with microwave-supported synthesis of silver nanoparticles/chitosan-grafted-polyvinyl alcohol hydrogels and their potential for wound healing alongside with antimicrobial activity. Although authors put a lot of effort into study and many experiments were performed, I could not recommend article for publication in presented form.

1.      The English presentation of whole work must be improved. A native English speaker should check the Manuscript.

2.      I do not see a point of usage “;” in many sentences. It is unnecessary.

3.      Authors should cite more relevant literature in the Introduction section. I’m afraid that only one citation after some statements is not enough. In my opinion, some paragraphs in the Introduction section should be rewritten or better organized.

4.      Please be uniform when you citing the literature according to journal instructions. For example, in Introduction you used numbers i.e., [1], while in some paragraphs the name of authors was used (line 143, etc.).

5.      The symbol for gram is g not gm. Also, be consistent with all units of measurement (h not hrs, etc.).

6.      Section 2.5. Were you used silver nitrate (AgNO3) or silver nanoparticles (Ag) when you prepared grafted PVA/AgNPs hydrogels? From presented text it is very confusing. I think you used silver nanoparticles as it can been seen from Figure 1, but in the text, you mentioned silver nitrate nanopowders.

7.      Lines 174 and 175. Authors said: “AgNPs (0.1 ml) were dispersed in 5 ml distilled; water.” Did you mean 0.1 g?

8.      It is very difficult to see AgNPs size range from obtained TEM micrographs. Please, provide size distribution charts or measure particles on micrographs.

9.      Line 265 and 266. Authors wrote: “The distribution of silver nanoparticles in S1Ag 0.8 hydrogel was higher than in S1Ag 0.3 hydrogel, according to UV-Vis data 266 (Figure 4)”. In figure 4 on y axes is wound size reduction rate, so I do not see a connection. Maybe, you put a wrong figure since the title also mismatches.

10.   What is a point of having TAgNPs1- TAgNPs4?

11.  At many places authors mentioned wrong Figure number. So, it is very difficult to follow.

12.  Authors wrote: “As shown in Figures 6 and 7, the quantity; of silver ions released; from S1Ag 0.3 and S1Ag0.8 hydrogels by utilizing; F12 medium with regard; to time was measured by AAS. The amount of silver; ions released from S1Ag 0.3 hydrogels was 5.9 g/ml within the first; 6 hours and quickly climbed to 13 g/ml over the next 7 hours. The release rate from 12 to 24 hours, reaching 18 g/ml. After; the first 24 hours, the release rate increased somewhat, reaching a value of 19 g/ml at 48 hours. In comparison to the findings from the S1Ag0.3 gel, the amount of silver released on the S1Ag0.8 gel was twice.” From Figure 5, I do not see this. Y scale is given in μg/ml while x scale has points to 300 h.

13.  “Definitely, at the same concentration of 100% extracted solution, polyvinyl; only had; about; 90% cell viability; while S1Ag0.3 and S1Ag0.8 had approximately 70% and 80% cell viability, respectively. Nevertheless, all samples cell viability was higher than 85%, which; was; the; threshold; between cytotoxicity and non-cytotoxicity, so they; can; be; considered noncytotoxic.” ??????

14.   Figure 6 is of very poor resolution while Figure 10 is incomplete since it is cropped.

15.  Where are the results of FT-IR spectroscopy?

16.  The time of 8 to 14 days is quite long. How do you know that natural processes are not responsible for wound healing?

Author Response

Dear Reviewer

Kindly find the attached file 

Thank you 

Reviewer 2 Report

Below, I send the comments on the manuscript presented by Fahad M. Aldakheel et al. 

1. Review the full article and avoid repetition of the symbol “;”

2. Line 19, delete extra “r” in silver word.

3. Line 26, replace “In” for “It”

4. Line 27, authors say: These composites were successfully used as antibacterial materials. Hence, they can be useful to help in healing an open wound. The current way in which it is written suggests that composites are already being used in clinical practice.

5. Please add the information about where the bacterial strains were obtained.

6. What was the reason the bacteria had to be identified? Were they clinical isolates? Please, justify it.

7. Names of microorganisms MUST be in italics.

8. Figure 2 should go in the results section. Please put it in the appropriate section. Also,

9. The aspect ratio of figure 2 is not appropriate. Please, fix it.

10. Indicate the catalog number of the cell line used.

11. Line 220, please indicate the cell density correctly, 105 (superscript 5) instead of 105.

12. Please, indicate the way in which the rats were sacrificed. We must remember that the guidelines on handling laboratory animals indicate that animal suffering must be minimized.

13. The legend of figure 4 is about the UV-Vis spectrum, which does not correspond to the graph presented. In addition, significant differences between groups are not indicated, which are described in the text.

14. Lines 278-279, authors say: Figures 5 and 6 of each sample for gram-negative strains and gram-positive strains. However, Figure 5 on page 8 does not correspond to what is said in the text.

15. Figure 6 should be substantially improved as the quality of the images is poor. Nothing can be concluded from them.

16. Lines 304-305, the authors say: all samples cell viability was higher than 85%, which was the threshold between cytotoxicity and non-cytotoxicity. Please, add the corresponding reference for this statement.

17. Line 313, Figure 8 MUST be substantially improved. The quality of this image is not acceptable for publication in any quality scientific journal.

18. Line 329, please verify that 23 2 to 30 9 (%) are correct. Same in line 331.

19. Line 333, Replace the word “subjects” with “animals”.

20. There is no statistical analysis in Figures 7 and 9.

Author Response

(The authors gave the same response as above.)

Reviewer 3 Report

the idea of the manuscript is repetitive

the manuscript lake the novelty

English language very poor

Author Response

Dear Reviewer

Kindly find the attached file 

I did the required modifications according to your advice 

Thank you 

Reviewer 4 Report

Journal: Molecules  

Title: Microwave-Supported Synthesis of Silver Nanoparticles/Chitosan-Grafted-Polyvinyl Alcohol Hydrogel for Wound Healing Applications

Green tea leaf extract was used as a natural reducing agent to synthesize silver nanoparticles; (AgNPs) using microwave irradiation technology for the reduction of silver ions to silver nanoparticles.

The paper sound and contributes to the field. In my opinion, it can be published after addressing the following points.

My comments:

1-    The article's grammar and punctuation are not adequate, and it needs to be deeply proofread.

2-     Figure 6 is almost unreadable due to its extremely poor quality. Please rebuild it.

3-     Figure 10: scale bars are not clear

4-    The introduction section needs a little improvement. I suggest the following very recent references in AgNPs: Sensors and Actuators A: Physical, Vol. 347, Article number 113942 (2022).  https://doi.org/10.1016/j.sna.2022.113942        and     Zeitschrift für Naturforschung A - A Journal of Physical Sciences, Vol. 77, No. 9, 909-919 (2022)  https://doi.org/10.1515/zna-2022-0126.

5-     The references need to be updated, please check and update the references

6-    Page 5: -Please indicate, what is LB.

7-    Please, mention the sizes of the AgNP within a Table. I could not find any size information resulting from the TEM.

8-      The discussion part needs to be arranged in a better way

Author Response

(The authors gave the same response as above.)

Round 2

Reviewer 1 Report

From the attached Author response, I could not find the response to my review. I repeated my decision, but if some mistake happened and authors provide response and make demanded changes, I will reconsider my decision.